# Nonsurgical Treatment Strategies for Elderly Head and Neck Cancer Patients: An Emerging Subject Worldwide

**DOI:** 10.3390/cancers14225689

**Published:** 2022-11-19

**Authors:** Hiroshi Okuda, Hirofumi Shibata, Takahiro Watanabe, Kosuke Terazawa, Kenichi Mori, Natsuko Ueda, Toshimitsu Ohashi, Takenori Ogawa

**Affiliations:** Department of Otolaryngology-Head and Neck Surgery, Gifu University Graduate School of Medicine, Gifu 501-1194, Japan

**Keywords:** elderly, head and neck squamous cell carcinoma, geriatric assessment, cisplatin, dose reduction, molecular-targeted drug, immune checkpoint inhibitor

## Abstract

**Simple Summary:**

Among elderly adults, there is an increasing rate of those receiving chemoradiotherapy to preserve function and maintain quality of life. Many currently used guidelines on therapies for head and neck squamous cell carcinoma (HNSCC) exclude elderly adults, although the number of elderly patients with HNSCC is drastically increasing worldwide. In this review, we present the current status and potential new treatment strategies for elderly patients with HNSCC. Poor tolerance to chemoradiotherapy-related toxicity is a bigger concern in older patients than in younger individuals. Then, we describe toxicities particularly severe in elderly patients and provide an overview of countermeasures in chemotherapy and radiotherapy, particularly with a focus on cisplatin-based chemotherapy (the main treatment for HNSCC). In addition, we describe the molecular target drug therapy and immunotherapy for older patients with HNSCC.

**Abstract:**

Consistent with the increasing rate of head and neck cancers among elderly adults, there has been an increase in the rate of those receiving nonsurgical treatments to maintain their function and quality of life. However, various problems, such as poor tolerance to chemoradiotherapy-related toxicity, are of greater concern in elderly adults than in younger individuals. In this review, we describe adverse events that should be particularly noted in elderly patients and provide an overview of countermeasures in nonsurgical treatments. We mainly focus on cisplatin-based chemoradiotherapy—the primary treatment for head and neck squamous cell carcinoma (HNSCC). Furthermore, we review the molecular targeted drugs and immune checkpoint inhibitors for elderly patients with HNSCC. Although the number of older patients is increasing worldwide, clinical trials aimed at determining the standard of care typically enroll younger or well-conditioned elderly patients. There is still very little evidence for treating elderly HNSCC older patients, and the question of optimal treatment needs to be explored.

## 1. Introduction

According to the United Nations, 703 million people (9% of the total population) are aged ≥65 years worldwide; this number is expected to increase by two times (1.5 billion; 16% of the total population) by 2050. Globally, aging is an emerging issue that may affect the number of patients with cancer.

Head and neck squamous cell carcinoma (HNSCC) is the sixth most common malignancy worldwide [1,2]. Notably, the annual proportion of elderly individuals with HNSCC has been increasing in recent years: approximately 25–40% of all patients with HNSCC are predicted to be elderly, with the trend expected to increase in the future [3,4,5,6]. As the population ages, treatment strategies for elderly HNSCC patients are becoming essential.

In general, elderly patients are defined as those aged ≥65 years [7]. According to a more detailed classification of the older adults, those aged 65–74, 75–84, 85–99, and ≥100 years are classified as pre-old, old, old-old, and oldest-old, respectively [7]. Further, different standards were used across different studies, and people aged ≥65 years were consistently classified as elderly or older individuals in some studies [8].

Regarding nonsurgical treatments, older individuals are more frail than younger individuals, and they often suffer from various complications [1,9,10], which sometimes reduce the effectiveness of the treatment. In addition, older adults often experience severe toxicity during treatment, forcing unexpected or early discontinuation of treatment [11]. Furthermore, late toxicities, such as dysphagia, persists long after treatment in older patients, which can easily lead to reduced quality of life, especially in cases of chemotherapy combined with radiotherapy [12].

Many well-known oncology clinical trials recruit relatively few elderly patients [13], and some ongoing trials continue to have upper age limits in their inclusion criteria [14]. In contrast, the general condition of the elderly patient varies from case to case, and the tolerability of chemotherapy can also vary in patients of the same age. Therefore, deciding chemotherapy options based on age alone may be associated with the risk of undermining the best treatment options for patients and may result in an unfavorable prognosis [15]. Recent advances in geriatric assessment may help risk stratification for nonsurgical treatments [16,17]. Because cisplatin still plays a central role in nonsurgical HNSCC treatments [18,19,20,21,22,23], it is important to first investigate how to use cisplatin in the elderly. In addition, recent advances in other chemotherapies, including molecularly targeted drugs and immunotherapy, have completely changed treatment strategies for HNSCC. However, many landmark studies have targeted young patients or elderly patients in good condition. It remains unclear whether there is enough evidence to establish treatment strategies for elderly patients with HNSCC.

In this review, we first focus on geriatric assessment for HNSCC, which supports treatment decisions for elderly patients. Subsequently, we review the current status of cisplatin-based definitive chemotherapy and radiotherapy, which may reduce toxicities and help manage older patients. Additionally, we provide new insights into treatments including molecularly targeted drugs and immune checkpoint inhibitors (ICIs) that are attracting attention.

## 2. Geriatric Assessment Tools for Cancer Therapy in Elderly

Older people tolerate chemotherapy poorly and are at increased risk of treatment toxicity compared with younger people [24]. However, excessive reduction in treatment intensity to prevent serious adverse events is detrimental to the patient.

Various tools have been developed to predict the toxicity of chemotherapy in the elderly. These models should be used as additional information that can help the decision-making process, along with clinical judgment and patient preference [17]. Appropriate pre-treatment toxicity prediction will help provide truly beneficial chemotherapy to the patient.

The comprehensive geriatric assessment (CGA) is an index used to accurately assess the tolerability of chemotherapy in elderly patients [16,25]. The CGA evaluates three aspects: life function, spirit and psychology, and society and environment. Specific evaluation items include ADL and activities of daily living, cognitive function, mood, emotion, happiness, motor function, urination function, communication ability, and social environment (home environment and care system). However, one concern of the CGA is the time it takes to evaluate all elements of the CGA. Geriatric 8 (G8) and the flemish version of the triage risk screening tool (fTRST), which are used as screening tools for elderly patients with cancer [25,26,27], can be completed in 5–10 min and are useful in determining treatment intensity. Furthermore, various attempts have been made to set cut-off values for these scores to utilize as prognostic factors. Neve et al. investigated G8 score before treatment in 35 elderly patients over 65 years old with head and neck cancer. They revealed that the group with G8 score of less than 14 had a trend toward lower treatment completion rates (75% vs. 100%) and longer mean hospital stays (12.2 vs. 6.5 days) [16]. Kenis et al. included 935 patients over 70 years old with various types of cancers and revealed that G8 score less than 14 or fTRST score over 1 were strongly prognostic factor for functional decline on ADL and IADL. Moreover, G8 score of less than 14, or fTRST score over 1 or 2 were significantly poor prognostic factors for OS. G8 score had the strongest prognostic value for OS (hazard ratio for G8 normal v abnormal, 0.38; 95% CI, 0.27 to 0.52) in these parameters [26]. Pottel et al. recruited 100 elderly patients over 65 years old with HNSCC. They demonstrated a significantly lower survival in patients with G8 score under 14 compared to patients with G8 score over 15 [27]. Ishii et al. investigated elderly patients with head and neck cancer over 65 years old and proposed G8 score cut-off value for overall survival was 10.5 (area under the curve (AUC) 0.69; 95% confidence interval (CI) 0.56–0.82). In their study, thirty-day mortality and all-complication rates were significantly higher in the low G8 score (<11) group [28]. These studies are summarized in Table 1 [4,16,26,27,28]. These data indicate that G8 score under 11–14 and fTRST over 1–2 are the cut-off values to distinguish high-risk elderly patients.

The cancer and aging research group (CARG) toxicity calculator is a pre-chemotherapy assessment that predicts the risk of developing grade 3 to 5 Common Terminology Criteria for Adverse Events (CTCAE) toxicity [29]. This tool refers to the following 11 items for evaluation. That is age, cancer type, planned chemotherapy dose, planned number of chemotherapy drugs, hemoglobin level, creatinine clearance level, hearing function, number of falls within 6 months, ability to self-administer oral medication, walking ability, and social activities. Hurria et al. suggested that this calculator demonstrated significantly better use than the clinician-assessed Karnofsky performance status (KPS) score at predicting toxicity of cancer treatment. In their study, the cohort was divided into three categories based on the risk of grade 3 to 5 toxicity: low risk (0 to 5 points), intermediate risk (6 to 9 points), and high risk (10 to 19 points). There was a significant difference in toxicity among the risk groups (*p* < 0.001). In comparison, there was no significant difference in the incidence of toxicity across the KPS-based risk groups [29].

Martine et al. proposed the chemotherapy risk assessment scale for high-age patients (CRASH) score. It was helpful to predict the risk of grade 4 hematological or grade 3 to 4 non-hematological toxicities [30]. This score is constituted by diastolic blood pressure, independent activities of daily living, lactate dehydrogenase level, Eastern Cooperative Oncology Group performance status (ECOG PS), mental status, nutritional status, and kinds of the regimen used. They classified these factors into 0, 1, or 2 points as each numerical value and revealed the higher points were associated with a higher risk of severe toxicity. These can be powerful tools for identifying high-risk elderly patients. Future studies must evaluate which tools are most reliable in prospective studies.

## 3. Cisplatin-Associated Toxicities

The anti-tumor effect of cisplatin occurs through DNA damage-induced apoptosis of tumor cells [31,32]. Importantly, tumor drug resistance, which may develop during chemotherapy, might prevent achieving sufficient therapeutic effects [33,34]. To overcome this critical issue, anti-tumor drugs with different mechanisms of action, such as taxanes, pyrimidines, and molecular targeted therapeutics, have been used in combination with cisplatin to treat HNSCC [32,35,36,37]. However, even now, cisplatin is the key drug for HNSCC nonsurgical treatment. In this paragraph below, we overview cisplatin-related toxicities and the management strategies.

### 3.1. Nephrotoxicity

Cisplatin-associated nephrotoxicity is primarily due to damage to the proximal tubules [38,39,40,41]. The metabolized cisplatin is excreted from the kidneys; thus, cisplatin accumulates in and is more likely to cause damage to the kidneys compared to other organs. About 30% of patients treated with cisplatin develop acute tubular necrosis as an acute toxicity and stromal fibrosis as late toxicity [38].

Renal function and morphology has been studied in rats, showing decreased number of tubular cells, tubular atrophy, and steatosis, along with other changes associated with aging [42]. Moreover, tubular length and volume can also be reduced markedly, and sparse areas of scarring, tubular atrophy, and tubular diverticula are common in the kidneys of elderly individuals [43]. This characteristic may cause cisplatin therapy in the elderly to cause more severe renal injury than in the younger patients. According to a previous meta-analysis by ZhiYu et al., the overall incidence of cisplatin-induced renal toxicity in elderly patients is 10.1% (121/1203), which is significantly higher than that in young patients (6.5%, 289/4439). This meta-analysis demonstrated the rate of renal toxicity by various chemotherapy including cisplatin. They revealed that the overall incidence of cisplatin-induced acute kidney injury (AKI) in elderly patients was 24.49% (9.5–48.6%), which is significantly higher than that in young patients (15.4%) [43]. In this latter analysis the author included studies of only a mono-therapy by cisplatin. Furthermore, the definition of AKI is more sensitive than the commonly used definition of renal toxicity, thus the rate of the toxicity was higher than that of the former meta-analysis.

Since no drugs are currently directly detoxifying cisplatin [39,40], hydration using large fluid volume together with a diuretic is widely used. Elderly patients are often at risk of heart failure due to excessive fluid load. Therefore, it is advisable to monitor for body weight gain and pleural effusion and to diligently use diuretics to prevent sudden changes in hemodynamics.

### 3.2. Myelosuppression

Oxidative stress and mutations in genes involved in hematopoiesis are the leading causes of cisplatin-induced myelosuppression [39,44,45,46]. Severe myelosuppression during treatment is reported in 5–6% of patients receiving cisplatin-containing chemotherapy for common solid tumors [47]. In elderly patients, the self-renewal ability and the number of hematopoietic stem cells in the bone marrow are reduced [48]. Thus, myelosuppression is more likely to be severe and prolonged in elderly patients. The risk of developing life-threatening complications such as febrile neutropenia is considered higher.

Very few reports focus on the difference in myelosuppression between younger and elderly patients. There may be some tendency to show higher severe myelosuppression rate in triweekly cisplatin treated group. However, most reports demonstrated only the rate of severe myelosuppression and median/mean age of participants [49,50,51,52,53,54,55,56,57,58,59], which made it difficult to clearly evaluate the effects of triweekly/weekly cisplatin for older patients (Table 2). It seems that the use of chemotherapy for the elderly will increase more and more in near future. Therefore, further research on the myelosuppression by chemo-therapy in the elderly is required.

In response to myelosuppression, rapid recovery can be expected following blood transfusion to address the reduction in red blood cells and platelets. For leukocytes and neutrophils, granulocyte colony-stimulating factor (G-CSF) administration is effective. Recently, studies have increasingly reported the cytoprotective effect of vanadium complexes in the bone marrow. In addition to G-CSF, these agents may serve as a promising candidates for reducing the adverse effects of cisplatin on bone marrow cells and cisplatin-related secondary complications such as febrile neutropenia in patients with cancer [44].

### 3.3. Nausea and Vomiting

The causes of nausea and vomiting caused by cisplatin are varied and include direct stimulation of the chemoreceptor trigger zone and the vomiting center, stimulation of the vomiting center by the vestibule or the gastrointestinal tract, and induction by psychological factors such as anxiety [60,61]. Especially in elderly patients, nausea and vomiting due to chemotherapy lead to weight loss and impose a reduction in treatment intensity, which is calculated based on the body surface area. Furthermore, malnutrition develops as a secondary adverse event. Body weight is one of the endpoints of the abovementioned G8 screening tool and an important factor in determining prognosis [25,26,27]. Arya et al. implemented a comparative study on chemoradiotherapy outcomes with cisplatin, carboplatin, and cetuximab in elderly patients with oropharyngeal squamous cell carcinoma. This study enrolled 409 patients 65 years old or older by applying data from the linked SEER-Medicare database. Patients with definitive chemoradiation with cisplatin required unplanned support for their severe nausea, emesis, or diarrhea in 33.5% of the total. This frequency was significantly high than that of patients who received chemoradiation with carboplatin and cetuximab (29.0% and 18.5%, respectively) [62]. Antiemetics targeting receptors involved in central and peripheral vomiting reflexes are used in cisplatin-treated patients experiencing nausea and vomiting. Several agents, including central and peripheral dopamine receptor antagonists (e.g., haloperidol and metoclopramide), 5-hydroxytryptamine type 3 receptor antagonists (e.g., granisetron and ondansetron), and neurokinin-1 receptor antagonists (e.g., aprepitant) are widely used, and studies have reported the efficacy of combination treatments [63,64,65]. Nonetheless, other underlying factors that cause nausea and vomiting, such as increased intracranial pressure and ileus, should not be overlooked.

### 3.4. Ototoxicity

Cisplatin induces apoptosis of cochlear hair cells, resulting in volume-dependent hearing impairment. Hearing deteriorates mainly in the treble range and is irreversible [66,67]. Tinnitus often develops with hearing loss, and patients may present with sleep disorders and depressive symptoms [68,69], especially in elderly patients. Hearing impairment is not life-threatening, but the poor quality of life can affect treatment. In fact, in one study, 14% of patients had difficulty completing treatment due to hearing loss during treatment for HNSCC [70]. In this study, 85 patients were administered high dose cisplatin except 2 patients started with carboplatin because of pre-existing neuropathy and hearing loss, respectively. It is unclear how many elderly patients could not complete the treatment specifically. However, we consider that elderly patients with age-related hearing impairment are more likely to suffer from ototoxicity of cisplatin and discontinue scheduled treatment. Therefore, elderly patients who require cisplatin treatment should be carefully managed.

Hearing impairment is irreversible, and management focuses on assistive devices such as hearing aids. Although acoustic therapy and counseling have also been effective, cognitive-behavioral therapy is currently the only proven treatment for tinnitus [69].

### 3.5. Other Toxicities

Cisplatin-induced oxidative stress and carnitine deficiency can lead to cardiac damage [71]. In addition, the hydration-induced cardiac load can lead to secondary cardiac disorders. Similarly, oxidative stress due to high doses of cisplatin results in hepatocellular injury [72]. These toxicities, albeit infrequent, can be fatal in elderly patients, and it is essential not to overlook abnormal blood test results or electrocardiographic changes.

## 4. Chemoradiation for Elderly Patients

In elderly patients, administering chemotherapy in combination with radiotherapy was challenging to the severe toxicity associated with radiotherapy. However, advances in radiotherapy methods, such as intensity-modulated radiotherapy and fractionated radiotherapy, have led to dramatic reductions in radiation-associated adverse events. As a result, it has become possible to use chemotherapy in combination with radiotherapy while minimizing associated toxicities [73,74,75,76]. The precise radiation methods appropriate for the elderly were written in elsewhere [1].

Radiation therapy is associated with acute and late toxicities. In radiation therapy for locally advanced HNSCC, typical acute toxicities include mucositis, dermatitis, dysphagia, decreased taste, and hoarseness. The late toxicities include radiation-induced osteonecrosis, xerostomia, subcutaneous fibroma, trismus, hypothyroidism, sensorineural hearing loss, and throat stenosis [77]. In a meta-analysis that included three RTOG trials evaluating the effect of concurrent chemoradiotherapy for locally advanced HNSCC, severe (G3–4) late toxicities rates were evaluated. A total of 230 patients were assessable, and 99 showed severe late toxicity. On multivariable analysis, older age (continuous value) was a significant variable (odds ratio: 3.07, *p* = 0.0036) [78]. These data suggest chemoradiotherapy for older patients may narrow second-line treatment options due to long-lasting late toxicities, reduce the quality of life, and increase the possibility of recurrence. Another meta-analysis that included 93 trials and 17,346 locally advanced HNSCC patients, examined the effects of adding chemotherapy to radiation therapy. In age-related analyzes, the treatment effect of adding chemotherapy was evident in those aged 70 years or younger, but the add-on effect diminished in those aged 71 years or older [13]. Based on these results, it is necessary to consider the efficacy and side effects of chemoradiotherapy when administering chemoradiotherapy to patients aged 71 years or older.

## 5. Appropriate Management of Chemotherapy in Elderly Patients

Few studies focused on the characteristic of chemoradiotherapy for the elderly [79,80,81]. Two retrospective studies indicated that elderly patients with HNSCC suffered worse complications such as myelosuppression and infections and were forced to more unplanned hospitalizations [79,81]. Mical et al. investigated 44 patients with HNSCC over 70 and 137 patients with HNSCC under 70 who received standard chemoradiotherapy. Total unplanned hospitalizations were required more often in the elderly, with at least one unexpected hospitalization occurring in 84% of the older while 67% of the younger patients (*p* = 0.031). Older patients experienced neutropenia more frequently than younger patients (*p* = 0.048) [64]. In a similar study by Merlano et al. (included 93 elderly and 224 younger, cut-off 65 years old), elderly patients suffered from infections, and in particular pneumonia (28.0% and 10.8%, respectively) more frequently than young patients (15.6% and 2.2%, respectively) (*p* = 0.017 and *p* = 0.002, respectively) [81]. These unfavorable events seemed to lead the prognosis worse. Interestingly, in these studies, one study demonstrated no significant difference as to overall survival (OS) between elderly and young groups [79,80], while the other two elderly groups demonstrated a worse prognosis as to OS [64,66]. Nguyen et al. reported that the 2-year OS rate was estimated to be 67.5% and 74% for the elderly and younger, respectively (*p* = 0.33). Mical et al. demonstrated that the 5-year OS rate and PFS rate of elderly versus younger were 49% vs. 63% for the 5-year OS rate and 69% vs. 71% for the 5-year PFS rate (*p* value was not available) [64]. Merlano et al. demonstrated that the total survival time elderly versus younger was 27.9 months vs. 45.7 months (*p* = 0.01, hazard ratio for death using the cox model: 1.51, 95% CI: 1.01–2.25) [81]. Furthermore, the elderly group in this study was able to receive less chemotherapy than that of the young group [81]. It is important to note that it is difficult to compare outcomes among different studies because many used different chemotherapy regimens within the same study. However, focusing on the detailed course of treatment during chemoradiotherapy in the elderly may help improve their long-term prognosis.

Continuous irradiation is a critical factor in successful chemoradiotherapy. Similarly, the therapeutic effect can be maximized with the systematic use of chemotherapy at a predetermined dose. Unplanned chemotherapy dose reduction or radiation therapy interruption allows the tumor to regrow and develop resistance to chemotherapy, leading to reduced local control and better tumor survival [82,83,84]. Treatment completion without treatment intensity reduction is an essential factor affecting prognosis. If severe toxicity during initial treatment requires dose reduction, prolonged toxicity may limit the choice of second-line treatment. Dickstein et al. reported the risk factor of radiation treatment interruptions for elderly patients over 70 years old [85]. In this study, they demonstrated that advanced stage had significantly higher odds of treatment interruption compared to early stage (OR: 2.64 [95% CI: 1.29–5.41], *p* = 0.008), and definitive chemoradiotherapy with induction was associated with greater odds of treatment interruptions (OR: 2.99 [95% CI: 1.03–8.63], *p* = 0.044). In terms of tumor site, hypopharynx was significantly associated with treatment interruption (OR: 5.6 [95% CI: 1.7–21.6], *p* = 0.006). Moreover, they revealed that patients who experienced a treatment interruption had significantly greater weight loss compared to those with continued treatment (8.9 ± 6.9% vs. 6.6 ± 5.7%, respectively, *p* = 0.042) and worse 3-year OS than those with uninterrupted treatment patients (62.5% vs. 70.1%, respectively, *p* = 0.01).

Therefore, choosing an appropriate administration method and controlling toxicity are critical to prevent dose reduction and radiation interruption during the initial treatment.

## 6. Cisplatin Administration: High-Dose Triweekly or Low-Dose Weekly

Chemoradiotherapy with cisplatin alone as definitive or postoperative adjuvant treatment is often performed as a primary treatment strategy. Cisplatin-containing regimens often require dose reduction, especially in elderly patients, mainly due to renal impairment. Carboplatin is often used instead of cisplatin in patients with severe renal impairment. However, extensive prospective studies demonstrating the comparability of carboplatin with cisplatin have not been performed. In addition, carboplatin has been reported to cause a higher rate of myelosuppression [86] and is not superior to cisplatin in terms of safety.

For definitive chemoradiotherapy, the currently recommended cisplatin dosing regimen is 100 mg/m^2^ every three weeks during irradiation [87]. This regimen focuses on the total dose of cisplatin and can be expected to have an excellent therapeutic effect, although concerns regarding severe toxicity remain. Another chemoradiotherapy regimen recommended for HNSCC includes the administration of cisplatin at 40 mg/m^2^ weekly during irradiation. Several clinical trials have compared the high-dose triweekly and low-dose weekly regimens (Table 3) [88,89,90,91,92]. Two large systematic reviews reported that the low-dose weekly regimen had a better toxicity profile, although the effect was equivalent to that of high-dose triweekly regimen [89,91]. In addition, based on these data, compared to weekly regimen, tri-weekly cisplatin regimen tended to involve severe myelosuppression. As mentioned above (Section 3.2), myelosuppression is more likely to be severe and prolonged in elderly patients, and the risk of developing life-threatening complications such as febrile neutropenia is considered higher. Therefore, although the clinical data is not insufficient, we consider that the low-dose weekly cisplatin regimen might be preferable in elderly patients. Of course, this possibility requires further prospective studies in elderly patients.

## 7. Molecular-Targeted Drugs

In a definitive therapy setting, the anti-Epidermal Growth Factor Receptor (EGFR) antibody (Cetuximab) had commonly used instead of cisplatin in elderly patients because of its perceived lower toxicity profile, as observed in the Bonner trial in definitive therapy [93]. In this study, cetuximab did not exacerbate the common toxic effects associated with head and neck radiotherapy, including mucositis, xerostomia, dysphagia, pain, weight loss, and performance-status deterioration. However, multiple retrospective studies are now published showing equal or increased incidence of toxicity in patients receiving cetuximab compared with patients receiving cisplatin concurrently with radiation [94,95]. Koutcher et al. reported that late Grade 3 or 4 toxicity was observed in 21 of 125 patients (16.8%) of the cisplatin-radiation with cisplatin group, while in 10 of 46 patients (21.7%) of the bioradiation with cetuximab group (*p* = 0.46). A similar report by Walsh et al. reported that 33 patients treated by radiation with cisplatin developed Grade 3 or 4 dermatitis in 6 patients (18%) and Grade 3 or 4 mucositis in 14 patients (42%) [93,94]. On the other hand, 34 patients who treated by radiation with cetuximab were developed Grade 3 or 4 dermatitis in 21 patients (62%) and Grade 3 or 4 mucositis in 24 patients (74%). These severe toxicities rates were significantly higher in cetuximab plus radiation therapy group (*p* = 0.0004 and 0.014, respectively [94,95]. In both these studies, elderly patients tended to be introduced to cetuximab rather than cisplatin due to decreased organ function, such as renal impairment. These data indicate that elderly patients without organ disabilities are recommended to use chemoradiation with cisplatin rather than bioradiation with cetuximab. In addition, in RTOG 1016 study, a randomized, non-inferiority clinical trial, radiotherapy plus cetuximab or cisplatin for human papillomavirus (HPV)-positive oropharyngeal cancer (OPC) were examined. In this trial, the radiotherapy plus cetuximab arm showed inferior OS and PFS compared with radiotherapy plus cisplatin arm in patients with HPV-positive OPC. The estimated 5-year OS was 77.9% (95% CI 73.4–82.5) in the cetuximab group versus 84.6% (80.6–88.6) in the cisplatin group. Estimated 5-year PFS was significantly lower in the cetuximab group (67.3%, 95% CI 62.4–72.2) compared with the cisplatin group (78.4%, 95% CI 73.8–83.0) (HR 1.72, 95% CI 1.29–2.29; *p* = 0.0002) [96]. Another randomized phase III study comparing chemoradiotherapy with cisplatin versus cetuximab in patients with locoregionally advanced HNSCC (ARTSCAN III) revealed that cetuximab is inferior to cisplatin regarding locoregional control for concomitant treatment with RT in patients with locoregionally advanced HNSCC [59]. At 3-year OS was 88% (95% CI, 83% to 94%) and 78% (95% CI, 71% to 85%) in the 145 cisplatin patients and 146 cetuximab patients, respectively (HR 1.63, 95% CI 0.93–2.86; *p* = 0.086). The cumulative incidence of locoregional failures at 3 years was 23% in the cetuximab group (95% CI 16–31%) compared with 9% (95% CI 4–14%) in the cisplatin group, which showed a significant difference (*p* = 0.0036). Although these two studies do not include elderly patients’ specific data, these data indicate that cetuximab in elderly HNSCC should be limited to patients with severe renal hypofunction or myelosuppression before treatment cisplatin-induced toxicity could involve fatal.

For recurrent and metastatic R/M HNSCC patients, combination therapies, including cetuximab, cisplatin, and fluorouracil (EXTREME regimen) [22], or cetuximab and paclitaxel is widely used. The EXTREME trial evaluated the effect of cetuximab on cisplatin/carboplatin plus 5-FU in R/M HNSCC. In total, 442 patients were enrolled, 17% of whom were older than 65 years. The primary endpoint was OS, with a median OS of 10.1 months in the total cetuximab-treated group and 7.4 months in the total control group. There was a significant prolonged OS in the cetuximab group (HR 0.80, *p* < 0.05), which underscored the clinical evidence of the Extreme regimen for R/M HNSCC. In contrast, in a subgroup analysis, there was a clear benefit of cetuximab for OS in patients < 65 years of age (HR 0.74) but no significant benefit in patients > 65 years of age (HR 1.07), highlighting the possible inferior effects of cetuximab in elderly patients.

## 8. Immune Checkpoint Inhibitors

Recent advances in tumor immunology led to the development of new agents that exert anti-tumor effects by a mechanism entirely different from conventional chemotherapy. With aging, a pro-inflammatory state is more likely to be shown as a result of a decrease in the number of naive CD4+/CD8+ T cells and in the repertoire of regulatory and memory T cells [97,98]. In addition, immuno-senescence also affects B-cell function, which may increase the risk of certain cancers, such as lymphoma [99]. Accurately assessing such immune-related problems unique to the elderly may be difficult at this time. However, immunotherapy with ICI treatments is believed to be better tolerated in the elderly than traditional cytotoxic anticancer agents.

HNSCC does not show dominant driver mutations that are amenable to intervention with molecularly targeted agents [100] and is a tumor type with a relatively high Tumor Mutational Burden (TMB) due to smoking, alcohol, and HPV infection. This evidence indicates that HNSCC is a tumor type likely to respond to immunotherapy [101]. The Checkmate 141 trial was a landmark clinical trial that not only demonstrated the efficacy of Immune Checkpoint Inhibitor (ICI) for R/M HNSCC but also the existence of anti-tumor immunity that can be stimulated in HNSCC with the use of anti-Programmed cell Death-1 (PD-1) antibody (Nivolumab). A total of 361 people participated in the trial, and 31% of patients were 65 years or older. The oldest patient in the nivolumab group was 83 years old. The primary endpoint was OS, with a median OS of 7.5 months in the nivolumab-treated group and 5.1 months in the control group, showing a significant prolongation in the nivolumab group (HR 0.70, *p* = 0.01) [102]. Notably, in subgroup analysis, OS was prolonged in the Nivolumab group even in patients aged 65 years or older, and no significant difference was observed in the incidence of side effects compared with those aged < 65 years [103]. These data indicate that anti-PD-1 antibody may be relatively safe to use in the elderly compared to other agents. The Keynote-048 trial also showed that the combination of anti-PD-1 (Pembrolizumab) and chemotherapy is effective against R/M HNSCC, and the efficacy depends on the combined positive score (CPS) of Programmed cell Death-1 Ligand-1 (PD-L1) expression by tumor and immune cells [104]. In the Keynote-048 trial, 882 patients were assigned to 3 arms (Pembrolizumab monotherapy, Pembrolizumab+ cisplatin/carboplatin+ 5-FU, EXTREME regimen). The median age of each group was 61–62 years. The pembrolizumab monotherapy group significantly prolonged OS compared with the EXTREME regimen at CPS ≥ 1 or CPS ≥ 20. Furthermore, the pembrolizumab plus chemotherapy group significantly prolonged OS compared to the EXTREME regimen in all patients. No subgroup analysis by age was reported in the Keynote-048 trial.

Elderly patients often cannot undergo surgery or standard chemoradiation therapy due to their general condition and comorbidities. Neoadjuvant immunotherapy has recently been used to treat HNSCC, and some studies have been reported, including elderly patients [105,106]. A multicenter phase II neoadjuvant immunotherapy trial which included elderly patients up to 87-year-old, reported no G3–4 severe adverse events in all patients, suggesting the clinical safety of this treatment. In neoadjuvant setting, tumor-specific T cells can effectively be activated in the presence of sufficient tumor-derived antigens and unaffected cervical lymph nodes from previous therapies such as chemotherapy, radiation, and surgery [107]. Neoadjuvant immunotherapies may also be an effective treatment for elderly patients who are not eligible for standard therapy. In addition to anti-PD-1 antibodies (Nivolumab, Pembrolizumab), other ICIs are also reported in the neoadjuvant setting, including anti-PD-L1 antibodies (atezolizumab, durvalumab), anti-Cytotoxic T-Lymphocyte Antigen 4 (CTLA-4) antibodies (ipilimumab, tremelimumab) [106,108], anti-Killer cell Immunoglobulin-like Receptor (KIR) antibody (Lirilumab) [109], and anti-T cell ImmunoGlobulin and ITIM domains (TIGIT) antibody (Tiragolumab), which are also likely to be eligible in elderly patients. Although most of these trials did not compare adverse event rates in older vs. younger patients, only 11% of G3 and none of the G4 adverse event rates were reported in a multicenter phase II trial including 85-year-old patients [109]. These data indicate the clinical safety of combinatorial neoadjuvant immunotherapy for elderly patients.

It is clear that adverse events related to ICIs differ from those related to conventional anticancer agents. For example, Anti-PD-1 antibodies exert anti-tumor effects by rejuvenating T cells within tumors or recruiting novel, antigen-specific T cells with anti-tumor activity from outside of the tumor [110,111]. Dermatological disorders, digestive disorders, and endocrine dysfunctions such as type 1 diabetes, thyroid dysfunction, and pituitary inflammation are frequently observed [112,113]. In ICI-treated patients with HNSCC, severe toxicity is generally low, with less than 1% grade 3 or higher severe toxicity reported in the Checkmate 141 trial [112].

Furthermore, clinical trials on ICIs often show a survival curve called the tail plateau, indicating that ICIs provide a certain degree of long-term therapeutic effect, and it is considered that the administration of ICIs reactivates the host’s immune system and induces long-term anti-tumor effects [114,115,116].

Low toxicity and promising long-term prognosis are significant advantages of ICI therapy in elderly patients [116,117]. In addition, an increasing number of patients who may have difficulty with conventional chemoradiotherapy due to poor tolerability of conventional chemotherapy may be considered to receive radiation therapy in combination with ICIs. Several clinical trials evaluate ICIs with radiation therapy in patients with HNSCC [118]. There is no statistical comparison of efficacy and adverse event rates in elderly vs. younger patients, and significant improvement of OS was not reported. However, the clinical safety of immunotherapy for elderly people is obvious and further research on ICIs may dramatically change the non-surgical treatment strategies in elderly patients with HNSCC.

## 9. Conclusions

In the present review, we provided evidence of appropriate nonsurgical approaches for elderly HNSCC patients, especially cisplatin-related treatments. In addition, we reviewed monoclonal targeted molecules and immune checkpoint inhibitors for elderly HNSCC patients. With the expected increase in average life expectancy and aging of patients with cancer in the future, first-line chemotherapy options defined by the currently used guidelines may not always be the best treatment for elderly patients. Further studies are needed to effectively assess the general condition and safely administer treatments that are truly appropriate for the patient.

## Figures and Tables

**Table 1 cancers-14-05689-t001:** Studies for geriatric assessment tools.

Author (Reference)	Article Type	Number of Patients	Study Population	Age Cutoff	Screening Tool	Cutoff for Difining as ‘Abnormal’
Szturz P et al. [4]	Systematic Review	N/A	Head and Neck	≥65	G8	≤14
					fTRST	≥1 or ≥2
Neve M et al. [16]	Pilot Study	35	Head and Neck	≥65	G8	≤14
Kenis C et al. [26]	Prospective Study	937	Various cancer types	≥70	G8	≤14
					fTRST	≥1 or ≥2
Pottel L et al. [27]	Prospective Study	51	Head and Neck	≥65	G8	≤14
Ishii R et al. [28]	Prospective Study	78	Head and Neck	≥65	G8	≤10.5

N/A: not available. G8: Geriatric 8 fTRST: flemish version of the triage risk screening tool.

**Table 2 cancers-14-05689-t002:** The rate of severe myelosuppression in definitive chemoradiotherapy with cisplatin.

Regimen					
triweekly cisplatin (100 mg/m^2^)					
Author (Reference)	*n*	age (median/mean)	toxicity Grade 3–5 (%)		
			neutropenia	anemia	thrombocytopenia
Wang Z et al. [49]	43	53	14	0	0
Ameri A et al. [50]	38	55.24	34.2	0	5.3
Visacri MB et al. [51]	29	56.6	6.9	17.2	0
Merlano MC et al. [52]	113	60	2	3	0
Kiyota N et al. [53]	129	61	49	14	3
Lim SH et al. [54]	19	61.7	0	0	0
Ahn D et al. [55]	150	62.2	16	6	1.3
weekly cisplatin (30–50 mg/m^2^)					
Author (Reference)	*n*	age (median/mean)	toxicity Grade 3–5 (%)		
			neutropenia	anemia	thrombocytopenia
Hamstra DA et al. [56]	29	47.7	3.4	3.4	0
Jacinto AA et al. [57]	20	53	5	0	0
Patil VM et al. [58]	268	54	3.4	1.5	1.5
Ameri A et al. [50]	39	55.46	17.9	2.6	5.1
Gebre-Medhin M et al. [59]	145	61	11	1	3
Kiyota N et al. [53]	122	62	35	13	4

**Table 3 cancers-14-05689-t003:** Studies comparing tri-weekly CDDP versus weekly CDDP.

Author (Reference)	Article Type	Number of Patients	Primary Site	Treatment	Age (Median)	Better Prognosis (Outcome)	Worse Toxicity (Events)
Szturz P et al. [88]	Systematic Review	4209	Head and Neck W/O Nasopharynx	definitive or postoperative CRT	N/A	Not significant (OS)	Tri-weekly (Myerosuppression, Naunea and Vomitting, Nephrotoxicity)
							Weekly (Dysphagia, Weight loss)
Bauml JM et al. [89]	Systematic Review	2901	Oral, Oropharynx, Hypopharynx, Larynx	definitive CRT	tri-weekly: 56–64 [60]	Not significant (OS)	Tri-weekly (Neutrosuppression, Nephrotoxicity, Ototoxicity, Dehydration, Electrolyte disturbance)
					weekly: 34–83 [61]		
Jacinto JK et al. [90]	Systematic Review	NA	Head and Neck	definitive or postoperative CRT	N/A	Not significant (OS, PFS)	Weekly (Mucositis)
Helfenstein S et al. [91]	Retrospective Study	314	Head and Neck	definitive or postoperative CRT	60	Not significant (OS, PFS)	Tri-weekly (Nephrotoxicity)
Geiger JL et al. [92]	Retrospective Study	104	Oropharynx	postoperative CRT	tri-weekly: 34–75 [53]	Not significant (OS, PFS)	Not significant
					weekly: 34–83 [61]		

CDDP: cisplatin, OS: overall survival, PFS: progression free survival, CRT: chemoradiotherapy, W/O: without, N/A: not available.

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
