# Peer review of "Nonsurgical Treatment Strategies for Elderly Head and Neck Cancer Patients: An Emerging Subject Worldwide"

_cancers, 2022, doi:10.3390/cancers14225689_

Round 1

Reviewer 1 Report

A beautiful and useful narrative review on an increasingly important topic. The main question addressed by the research is an interesting narrative review on non-surgical treatment strategies for elderly head and neck cancer patients. Many currently used guidelines on  squamous cell carcinoma therapies exclude elderly patients and elderly HNSCC patients are drastically increasing worldwide.

This is a recent review that takes into account every aspect of non-surgical treatment, from radiotherapy to recent immunotherapy. The conclusions are consistent with the evidence and arguments presented and they addressed the main question posed. The references are appropriate.

The major limit is the small number of data in the literature. Three simple tables that do not help to understand the text. The tables summarize the references.

Furthermore, the research methodology is not defined. Research is not systematic. The PRISMA scheme is absent. It does not follow the PRISMA guidelines. Search terms are not describe.

Author Response

Reviewer #1
A beautiful and useful narrative review on an increasingly important topic. The main question addressed by the research is an interesting narrative review on non-surgical treatment strategies for elderly head and neck cancer patients. Many currently used guidelines on  squamous cell carcinoma therapies exclude elderly patients and elderly HNSCC patients are drastically increasing worldwide.

This is a recent review that takes into account every aspect of non-surgical treatment, from radiotherapy to recent immunotherapy. The conclusions are consistent with the evidence and arguments presented and they addressed the main question posed. The references are appropriate.

The major limit is the small number of data in the literature. Three simple tables that do not help to understand the text. The tables summarize the references.

Furthermore, the research methodology is not defined. Research is not systematic. The PRISMA scheme is absent. It does not follow the PRISMA guidelines. Search terms are not describe.

We really appreciate the Reviewer #1 for the insightful, suggestive comments. As the Reviewer #1 writes, this is a narrative review, not a systematic review nor meta-analysis. In addition, there are very little evidence which directly compared non-surgical treatments in elderly vs. younger HNSCC patients. As most of clinical studies include young patients or elderly in good condition, which make it difficult to extract impactful statistical analysis related to elderly patients in severe condition. As such, although we know we have to write the manuscript more systematically, we couldn’t attach the PRISMA scheme. We appreciate the comments from the Reviewer #1. After the revise, we convince our manuscript has greatly improved.

Reviewer 2 Report

Thank you for submitting this article for review.   

There are some areas that require more clarification

*p6 3.4 Ototoxicity - please include specific data on ototoxicity in elderly patients vs. others  ?how much more risk of cisplatin induced ototoxicity for age >65; is this only for high dose cisplatin? or for weekly cisplatin too?

*p 7  lines 294-302 - please provide specific details on treatment interruptions in elderly?  What is the data on radiation treatment interruptions in patients > 65?

*p8 - line 321 - please expand on why "low dose cisplatin...preferable in elderly"   What is the data?  From the articles listed in Table 3, can the authors extract the data for elderly patients?  and give a summary of tri-weekly vs. weekly cisplatin toxicity risk in elderly patients vs. younger patients?

*page 9 - line 378 - 381 - would leave this paragraph out as it doesn't add information for elderly patients

*page 10 line 417 - 442 please give specific data on neoadjuvant immunotherapy side effects in elderly?  what is toxicity vs. younger patients?

*page 10 - line 446 - "clinical trials evaluate ICIs with radiation therapy"  are elderly patients included in these trials?  what is the enrollment for elderly patients expected in these trials?

Minor - several places have 2 decimal places - ie page 5 line 212  33.53% - only one decimal place would suffice

thank you for the opportunity to review your paper.

Author Response

Reviewer #2

Thank you for submitting this article for review.   

There are some areas that require more clarification

*p6 3.4 Ototoxicity - please include specific data on ototoxicity in elderly patients vs. others  ?how much more risk of cisplatin induced ototoxicity for age >65; is this only for high dose cisplatin? or for weekly cisplatin too?

Thank you for the insightful comment. In the indicated study (Reference [70]), 85 patients were received high dose cisplatin except 2 patients started with carboplatin because of pre-existing neuropathy and hearing loss, respectively. It is unclear that how many elderly patients could not complete the treatment. However, we would emphasize that hearing loss affects severely the patient’s quality of life, thus we consider that elderly patients with age-related hearing impairment are more likely to suffer from ototoxicity of cisplatin and discontinue scheduled treatment.

We have corrected and added the sentence to the indicated section (page 6, line 237-243).

*p 7  lines 294-302 - please provide specific details on treatment interruptions in elderly?  What is the data on radiation treatment interruptions in patients > 65?

We thank the Reviewer #2 for careful observation. We added a previous report with specific details on treatment interruptions in elderly, especially as to radiation treatment interruptions. In the study the authors suggested that the risk of radiation interruptions in the elderly were associated with the clinical stages, treatment methods, and tumor sites. Moreover, they revealed that the elderly patients who experienced a treatment interruption had significantly greater weight loss and worse overall survival.

We added the sentences and reference as mentioned above (p8, line 313-326).

*p8 - line 321 - please expand on why "low dose cisplatin...preferable in elderly"   What is the data?  From the articles listed in Table 3, can the authors extract the data for elderly patients?  and give a summary of tri-weekly vs. weekly cisplatin toxicity risk in elderly patients vs. younger patients?

Thank you for the critical comment. We searched the related article again and investigated the details carefully. We could not extract data focusing on elderly patients only. However, compared to weekly regimen, tri-weekly cisplatin regimen more tended to involve myelosuppression in two large systematic review. In section 3, we mentioned that myelosuppression is more likely to be severe and prolonged in elderly patients and the risk of developing life-threatening complications such as febrile neutropenia is considered higher. Therefore, we consider that low-dose weekly cisplatin regimen might be preferable in elderly patients.

We have added the summary of tri-weekly vs. weekly cisplatin toxicity risk, and corrected the sentences to explain why low-dose weekly cisplatin regimen might be preferable in elderly patients (page 8, line 342-351).

*page 9 - line 378 - 381 - would leave this paragraph out as it doesn't add information for elderly patients

Thank you for the striking comment. According to the insightful indication, we have left this paragraph out and deleted (page 10).

*page 10 line 417 - 442 please give specific data on neoadjuvant immunotherapy side effects in elderly?  what is toxicity vs. younger patients?

We appreciate the great comment from Reviewer #2. Most of these trials included elderly patients, but they didn’t statistically compare the toxicity rate in older vs. younger patients. However, some of these studies reported no or low G3 severe adverse events in all patients including elderly patients, suggesting the clinical safety of these treatments.

We have rewritten the sentences in page 11, line 445-447

We added sentences on page 11. Line__.*page 10 - line 446 - "clinical trials evaluate ICIs with radiation therapy"  are elderly patients included in these trials?  what is the enrollment for elderly patients expected in these trials?

Thank you for the comments. We added the sentences below on page 11, line 457-460.

There is no statistical comparison of efficacy and adverse event rates in elderly vs. younger patients, and significant improvement of OS was not reported. However, the clinical safety of immunotherapy for elderly people is obvious and further research on ICIs may dramatically change the non-surgical treatment strategies in elderly patients with HNSCC.

Minor - several places have 2 decimal places - ie page 5 line 212  33.53% - only one decimal place would suffice

We thank the Reviewer for their careful observation. We have made corrections to the decimal places as your suggestion.

thank you for the opportunity to review your paper.

We really appreciate the Reviewer #2 for these insightful comments. We convince our manuscript has greatly improved after revise.

Reviewer 3 Report

The manuscript is a review of the management of head and neck cancer chemoradiation, focusing on management of elderly patients. Such patients are under-represented in clinical trials.  The authors cite some literature that includes such patients, and in other cases extrapolate or make theoretical arguments.  This is an important topic, and the authors have put together a reasonable discussion of the topic.

The English in the abstract and introduction needs a lot of work. I did not correct most of it. The English improves starting on page 3.

1) abstract, line 29. remove the word "drastically"

2) line 45.  Change "The more older people increase worldwide..." to "As the population ages, treatment strategies for elderly NHSCC patients are becoming essential."

3) line 156.  Sentence unclear, unclear what is meant by "blank control group".  I believe the authors are trying to state that aging can lead to renal tubule atrophy, steatosis, and renal senescence, and cite animal data in reference 42.  Sentence needs to be rewritten.

4) line 161-166- numbers given appear to be from the same reference (43), but they are inconsistent. Overall incidence of renal toxicity in elderly patients is 10.6%... then they say overall incidence of acute renal injury in elderly patients is 24.49%...    How can the overall renal toxicity be less than the acute toxicity?

5) line 187-188, and Table 2.  The authors claim that table 2 shows a correlation between severe neutropenia and age.  However, only the mean or median age is available from the studies cited, and there is no credible correlation with toxicity.  If there is a real correlation, perhaps a graph would make it clearer.

6) line 288.  Change "administered little cycle of" to "able to receive less"

7) Line 294.  Change "the most" to "a"

8) Line 421.  "unaffected cervical lymph nodes from previous therapy".  Unclear what the term "previous therapy" is referring to. Is this prior immunotherapy (the neoadjuvant immunotherapy), or some other therapy?

Author Response

Reviewer #3

The manuscript is a review of the management of head and neck cancer chemoradiation, focusing on management of elderly patients. Such patients are under-represented in clinical trials.  The authors cite some literature that includes such patients, and in other cases extrapolate or make theoretical arguments.  This is an important topic, and the authors have put together a reasonable discussion of the topic.

The English in the abstract and introduction needs a lot of work. I did not correct most of it. The English improves starting on page 3.

We thank the Reviewer for their appropriate suggestion. We asked a native English-speaking grammar proofreader to proofread the manuscript on pages 1-2 (short summary, abstract, introduction) again and corrected the entire sentences. The English edited area was highlighted.

1) abstract, line 29. remove the word "drastically"

We thank the Reviewer for their careful observation. We have made corrections to the indicated words.

2) line 45.  Change "The more older people increase worldwide..." to "As the population ages, treatment strategies for elderly NHSCC patients are becoming essential."

We thank the Reviewer #3 for their careful observation. We have made corrections to the indicated words (page 2, line 45-46).

3) line 156.  Sentence unclear, unclear what is meant by "blank control group".  I believe the authors are trying to state that aging can lead to renal tubule atrophy, steatosis, and renal senescence, and cite animal data in reference 42.  Sentence needs to be rewritten.

We apologize for unclear and insufficient description. The aim of this study (reference [42]) was to examine the effect of short-term calorie restriction on cisplatin-induced nephrotoxicity in aged rats. In this study, aged rats with no intervention demonstrated renal tubule atrophy, steatosis, and renal senescence.

We have rewritten the indicated sentence (page 4, line 158-159).

4) line 161-166- numbers given appear to be from the same reference (43), but they are inconsistent. Overall incidence of renal toxicity in elderly patients is 10.6%... then they say overall incidence of acute renal injury in elderly patients is 24.49%...    How can the overall renal toxicity be less than the acute toxicity?

We apologize for unclear and insufficient description. The former meta-analysis demonstrated the rare of renal toxicity by various chemotherapy including cisplatin. On the other hand, in the latter analysis the author recruited studies of only a mono-therapy by cisplatin. Furthermore, in the latter analysis the definition of acute renal injury is more sensitive than the commonly used definition of renal toxicity.

We have added the sentence resolving the inconsistency in the indicated section (page 4, line 165-166, line 169-172).

5) line 187-188, and Table 2.  The authors claim that table 2 shows a correlation between severe neutropenia and age.  However, only the mean or median age is available from the studies cited, and there is no credible correlation with toxicity.  If there is a real correlation, perhaps a graph would make it clearer.

We thank the Reviewer #3 for their insightful comments. As your indication, we could not reveal significant correlation between severe neutropenia and age.

There may be some tendency to show higher severe myelosuppression rate in triweekly cisplatin treated group. However, most reports demonstrated only the rate of severe myelosuppression and median/mean age of participants, which made it difficult to clearly evaluate the effects of triweekly/weekly cisplatin for older patients. It seems that the use of chemotherapy for the elderly will increase more and more in the future. Therefore, further research on the myelosuppression of chemotherapy in the elderly is required

In this section, we have rewritten the text only to introduce several reports of chemoradiation-induced myelosuppression in HNSCC and to mention the need for further studies focused on the elderly. (page 5, 186-193).

6) line 288.  Change "administered little cycle of" to "able to receive less"

We thank the Reviewer for their careful observation. We have made corrections to the indicated words (page 8, line 300).

7) Line 294.  Change "the most" to "a"

Thank you for the great comment. We have made corrections to the indicated words (page 8, line 306).

8) Line 421.  "unaffected cervical lymph nodes from previous therapy".  Unclear what the term "previous therapy" is referring to. Is this prior immunotherapy (the neoadjuvant immunotherapy), or some other therapy?

We apologize for unclear description. The term "previous therapy" includes various therapies affecting the activation of lymph nodes such as chemotherapy, radiation, and surgery.

We have added the sentence to the indicated term (page 11, line 449).

We really appreciate the Reviewer #3 for these useful comments. We convince our manuscript has greatly improved after revise.

Round 2

Reviewer 2 Report

Thank you for your revisions.  Final decision on your manuscript will be communicated after editor review.

Author Response

We thank Reviewer #2 for the great comments. We convince our manuscript has improved after the revision.

Reviewer 3 Report

Prior  suggestions were carried out reasonably well.  English still needs to be carefully corrected.  There are many cases where the words "a" or "the" are omitted.  

Some specific corrections: 

abstradt line 29: eliminate the word "drastically"

line 60, introduction:  Modify sentence to: Many well-known oncology clinical trials recruit relatively few elderly patients (13), and some...  

Line 158, section 3.1.  Change first sentence to: Renal function and morphology has been studied in rats, showing decreased number of tubular cells, tubular atrophy, and steatosis, along with other changes associated with aging (42). 

line 169, section 3.1.  Change "In this analysis... " to "In this latter analysis..."

line 453: change "duravalumab" to durvalumab.

Author Response

We express great gratitude for the careful observation of Reviewer #3. We have checked and corrected some sentences suggested by Reviewer #3. Furthermore, we checked articles (a/the) as much as we could. We really appreciate Reviewer #3 for these valuable comments. We convince our manuscript has dramatically improved after the revision.